# Dyadic Interactions of Treatment-Resistant Schizophrenia Patients Having Followed Virtual Reality Therapy: A Content Analysis

**DOI:** 10.3390/jcm12062299

**Published:** 2023-03-15

**Authors:** Alexandre Hudon, Jonathan Couture, Laura Dellazizzo, Mélissa Beaudoin, Kingsada Phraxayavong, Stéphane Potvin, Alexandre Dumais

**Affiliations:** 1Centre de Recherche de l’Institut Universitaire en Santé Mentale de Montréal, Montréal, QC H1N 3J4, Canada; 2Department of Psychiatry and Addictology, Faculty of Medicine, Université de Montréal, Montréal, QC H3T 1J4, Canada; 3Faculty of Medicine, Department of Medicine, Campus Montréal, Université de Montréal, Montréal, QC H3T 1J4, Canada; 4Services et Recherches Psychiatriques AD, Montréal, QC H1C 1H1, Canada; 5Institut National de Psychiatrie Légale Philippe-Pinel, Montréal, QC H1C 1H1, Canada

**Keywords:** psychotherapy, virtual reality therapy, auditory hallucinations, schizophrenia, avatar therapy, dyads, dyadic relationship

## Abstract

(1) Background: Very little is known about the inner therapeutic processes of psychotherapy interventions for patients suffering from treatment-resistant schizophrenia. Avatar therapy (AT) is one such modalities in which the patient is undergoing immersive sessions in which they interact with an Avatar representing their main persistent auditory verbal hallucination. The aim of this study is to identify the most prevalent dyadic interactions between the patient and the Avatar in AT for patient’s suffering from TRS. (2) Methods: A content analysis of 256 verbatims originating from 32 patients who completed AT between 2017 and 2022 at the Institut universitaire en santé mentale de Montréal was conducted to identify dyadic interactions between the patients and their Avatar. (3) Results: Five key dyads were identified to occur on average more than 10 times for each participant during the immersive sessions across their AT: (Avatar: Reinforcement, Patient: Self-affirmation), (Avatar: Provocation, Patient: Self-affirmation), (Avatar: Coping mechanisms, Patient: Prevention), (Patient: Self-affirmation, Avatar: Reinforcement), and (Patient: Self-appraisal, Avatar: Reinforcement). (4) Conclusion: These dyads offer a first qualitative insight to the interpersonal dynamics and patient-avatar relationships taking place during AT. Future studies on the implication of such dyadic interactions with the therapeutic outcome of AT should be conducted considering the importance of dyadic relationships in psychotherapy.

## 1. Introduction

Psychotherapy is a complementary treatment to medication for many psychopathologies in mental health [1,2]. Individual psychotherapy is defined as an approach in which a therapist and patient are interacting together to improve psychopathologic conditions and functional impairment through the therapeutic alliance [3]. The mechanisms establishing the success of a psychotherapy are widely debated [2,4]. Despite the debates, the therapeutic bond between the therapist and the patient is perceived to be a major factor resulting in behavioral, social, or cognitive changes [5,6]. As part of this bond, the notion of transference and countertransference is relevant. Derived from psychodynamic paradigms, transference is defined as a process in which individuals displace patterns of behavior that evolve through interaction with significant figures in childhood onto other persons in their current lives. On the other hand, countertransference is known to be a corresponding response of the therapist following the transference [7,8,9]. These notions have clinical implications as they provide insight to the inner world of all parties involved in the therapeutic process [10,11]. For example, regarding transference in classical cognitive-behavioral therapy (CBT), it has been suggested that therapists should not deliberately provoke or ignore their patient; instead, they should be aware of their own feeling and monitor them [12]. Psychotherapeutic approaches can also be used for complex mental illness as an adjunct to psychopharmaceutical recommendations. Notably, a 25-year systematic review and exploratory meta-analysis reported that CBT was the most frequently recommended psychotherapy intervention for patients suffering from treatment-resistant schizophrenia (TRS) [13].

As stated above, one example of severe and complex mental illness is TRS. Schizophrenia is part of the psychotic disorders and is characterized by the presence of positive symptoms and negative symptoms, with an occupational or social dysfunction, with continued and persistent disturbance for more than 6 months [14]. Frequently seen positive symptoms are hallucinations (mostly auditory) and delusions, both of which are hypothesized to be due to hyperactive dopaminergic activation in the mesolimbic system [15,16]. Negative symptoms consist of five constructs: blunted affect, asociality, anhedonia, alogia, and avolition [17]. While various definitions exist for TRS, one that is widely used is a documented failure to two or more antipsychotics [18]. Around 20–30% of patients suffering from schizophrenia will evolve to TRS and about 40–70% of these patients will not respond to the treatment of choice (Clozapine) [19]. Therefore, there was a crucial need to identify new treatment approaches for these patients [19,20,21].

Several studies demonstrated benefits associated with psychotherapeutic approaches for targeting the positive symptoms of patients suffering from TRS [22,23]. Such techniques include CBT designed for psychosis. However, considering the mitigated results, further techniques were developed, such as avatar therapy (AT) [24,25]. This immersive therapeutical approach was developed by Julian Leff in 2008 [26]. In AT, the patients interact with a virtual representation of the patient’s most disturbing auditory verbal persistent hallucination, which we refer to as ‘’the Avatar’’. Pilot studies investigating the effects of AT demonstrated an improvement in patients suffering from TRS by reducing their auditory hallucination with a large effect size and positive changes in their quality of life [27,28,29]. Moreover, attempts were made to further understand the intrinsic psychotherapeutic processes. A first qualitative analysis exploring the dialogue components of AT identified five major themes amongst the patients: emotional responses to the voices, beliefs about voices and schizophrenia, self-perceptions, coping mechanisms, and aspirations [30]. These results were further developed by Beaudoin and her team by conducting a content analysis of the verbatims of 18 patients who received AT. In doing so, they were able to sub-divide the previously identified themes as well as explore the themes emerging from the avatar’s interactions [31]. Provocation, mainly the act of provoking the patient intentionally to elicit a reaction, was identified to be one of the most frequent interactions. However, the tuple (combination of two interactions) of interactions between the avatar and the patients have never been explored together as unique units of interactions known as dyads. The consideration of this combination is relevant in order to better understand the inner processes of AT considering that, in comparison with CBT, the therapist takes a more active role in coaching the patient as to how to respond to their auditory hallucinations. This may influence how the patient will respond to their visual representation of their auditory hallucinations. This is important, as it reports to the notion of transference and countertransference and its integration across a virtual environment. Furthermore, it has been previously observed that social interactions during psychotherapeutic interactions in virtual reality can be explored in different psychopathologies, such as in social anxiety. The engagement itself between the patient and a virtual human demonstrated a reduction in social anxiety in 18 participants suffering from high levels of social anxiety [32].

The aim of this study is to identify the most prevalent dyadic interactions between the patient and the Avatar in AT for patients suffering from TRS. It is hypothesized that certain dyadic interactions, such as the ones implying a provocation from the Avatar and a counterattack from the patient, will be predominant considering the importance of this theme in the therapy. Considering the immersive nature of this therapeutic approach, the dyadic interventions could provide additional understanding of the psychotherapeutic processes of AT to further explore the nature of transference and countertransference elements of this therapeutic approach.

## 2. Materials and Methods

### 2.1. Participants and Recruitment

The data used in this study were derived from participants who received AT in the context of a pilot clinical trial as well as one ongoing trial comparing AT to CBT, all conducted at the Centre de recherche de l’Institut universitaire en santé mentale de Montréal (CR-IUSMM) [28,29]. The participants all belong to the clinical trials registered on Clinicaltrials.gov (identifier numbers: NCT03585127 and NCT04054778). The delivery of AT was the same across the two trials as protocolized. Participants in these studies received nine one-hour psychotherapeutic sessions, of which eight were immersive sessions. The patients received AT from one of the only two therapists trained to provide AT at CR-IUSMM. During these sessions, they interacted with a virtual representation of their auditory verbal hallucinations: the Avatar. The participants included in this study were all patients at the IUSMM, above 18 years of age, and suffering from TRS as defined by the absence of response to two or more dopaminergic antagonists and receiving AT between 2017 and 2022. Study has been approved by the ethics committee of CR-IUSMM as part of the protocol for AT.

### 2.2. Data Collection

The immersive sessions of 32 patients who underwent AT were transcribed to verbatims from audio recordings by research auxiliaries. The verbatims were then verified by AH to ensure integrity of the transcriptions. This yielded 256 verbatims representing over 230 h of immersion in AT. Annotations of the interactions between the patients and the avatars were classified as per the 27 themes described in Beaudoin et al. [31]. The themes are presented in Table 1 for the Avatar and Table 2 for the patients.

The annotation of the verbatims was done automatically by using a peer-reviewed trained linear support vector classifier, previously trained and implemented on a dataset for AT using Python 3.9 with the Scitkit-Learn open library and a 10-k fold cross validation [33].

The performance for these initial automated annotations for the Avatar and the patients ‘themes is presented in Table 3. The results are comparable to the ones obtained in the description of the original study on automated classification of interaction for AT [33].

### 2.3. Data Analysis

#### 2.3.1. Dyadic Interactions

A dyad is the combination of two successive themes that results from the interaction of the Avatar with the patient or vice-versa. It is, therefore, the result of the engagement between the patient and the Avatar over two consecutive interactions.

For example, an (Avatar, Patient) dyad could be represented as follows:

(Avatar: Accusations, Patient: Self-affirmation).

Conversely, a (Patient, Avatar) dyad could be represented as follows:

(Patient: Prevention, Avatar: Beliefs).

#### 2.3.2. Analysis of Dyads

Using a Python script developed by A.H., dyads of interactions between the patients and their Avatar were compiled into an Excel spreadsheet and the frequencies of apparition of each dyad per verbatim were counted. The mean frequencies of each dyad for each participant were computed.

Dyads of interactions between the therapists were compiled into an Excel spreadsheet using a Python script developed by A.H. and was compared manually for four patients by J.C. The frequencies of apparition of each dyad per verbatim were counted. The mean frequencies of each dyad for each participant were computed. Dyads were selected for this study if they had a mean frequency above 10. The mean frequency above 10 was selected as most of the dyads had a mean frequency of four and below in the descriptive statistical observations of the identified dyads.

## 3. Results

### 3.1. Sample Characteristics

From the 256 verbatims of the 32 patients who underwent AT, a total of 1117 dyads were identified. Patients’ characteristics are presented in Table 4. Out of the identified dyadic interactions, only five dyads presented a mean frequency above 10. Figure 1 displays the identified dyads as well as their mean frequencies.

### 3.2. Dyadic Interactions

#### 3.2.1. Avatar: Reinforcement, Patient: Self-Affirmation

The dyadic interaction unit with the highest mean frequency is (Avatar: Reinforcement, Patient: Self-affirmation). It is characterized by an attempt of the Avatar to reinforce a statement made by the patient followed by a self-affirmation statement expressed by the patient. Such circumstances can be noted when the Avatar specifies an action that was completed by the patient to elicit further affirmations from the patient regarding this action.

For example, in verbatim 21005—T4:

Avatar 21005 (Reinforcement): “That’s it. You can tell me directly like this if you have something to tell me”.

Patient 21005 (Self-affirmation): “Well, it’s just that I felt badly when you elevated the tone of your voice. You seemed angry and it affected me”.

#### 3.2.2. Avatar: Provocation, Patient: Self-Affirmation

A provocation expressed by the Avatar followed by a patient’s self-affirmation yields another frequently identified dyadic interaction: (Avatar: Provocation, Patient: Self-affirmation). Provocations are frequent in AT and this is often performed by the Avatar to trigger an emotional response from the patient.

An example is found in verbatim 004-T5:

Avatar 004 (Provocation): “You are just someone who is not convincing at all”.

Patient 004 (Self-affirmation): “But if you think about it, maybe I should be changing something because I recognize that it does not make sense”.

#### 3.2.3. Avatar: Coping Mechanisms, Patient: Prevention

Coping mechanisms as per the Avatar’s theme is often under the form of a question related to the behavior of the patient or cognitive processes regarding a particular response. The dyadic interaction (Avatar: Coping mechanisms, Patient: Prevention) identifies a question (as stated above) followed by a response from the patient that states a prevention strategy they believe will help them deal with their auditory hallucinations.

This is illustrated in verbatim 008-T4:

Avatar 008 (Coping mechanisms): “How would you want me to leave your brain?”

Patient 008 (Prevention): “I think I’ll just stop listening. This way you will leave my brain”.

#### 3.2.4. Patient: Self-Affirmation, Avatar: Reinforcement

While the (Avatar: Reinforcement, Patient: Self-affirmation) dyadic interaction was the most prevalent in terms of its mean frequency, its reverse was also identified substantially. The (Patient: Self-affirmation, Avatar: Reinforcement) dyad implies a positive assertion from the patient that demonstrates some self-confidence, followed by a further assertion from the Avatar to encourage more positive assertions.

Such an occurrence can be identified in verbatim 1015-T5:

Patient 1015 (Self-affirmation): “I believe this is right. It changed something for me”.

Avatar 1015 (Reinforcement): “You seem very decided”.

#### 3.2.5. Patient: Self-Appraisal, Avatar: Reinforcement

When the patient compliments themselves or agrees with a compliment made by the Avatar and this is followed by a positive assertion from the Avatar, the dyadic interaction (Patient: Self-appraisal, Avatar: Reinforcement) is observed. This is often seen when the Avatar attempts to validate a positive statement made by the patient to elicit a further assertion or emotional response.

An example is found in verbatim 1011-T4:

Patient 1011 (Self-appraisal): “I am very kind and respectful!”

Avatar 1011 (Reinforcement): “Oh! You are starting to affirm yourself!”

## 4. Discussion

The main objective of this study was to identify the dyadic interactions between the patient and the Avatar that occur the most frequently in AT. The in-depth analysis of 256 verbatims from 32 patients were automatically annotated and the different types of interactions were compiled into dyadic interactions for further analysis. Amongst the identified dyads, five dyadic interactions occurred more than 10 times on average per participant. These dyadic interactions are, in decreasing order of prevalence: (Avatar: Reinforcement, Patient: Self-affirmation), (Avatar: Provocation, Patient: Self-affirmation), (Avatar: Coping mechanisms, Patient: Prevention), (Patient: Self-affirmation, Avatar: Reinforcement), and (Patient: Self-appraisal, Avatar: Reinforcement).

Current studies of psychotherapeutic processes urge the consideration of dyadic interactions to better understand co-regulation, receptiveness, and influence in interpersonal relationships [34]. During AT, the patients and the Avatar (animated by the therapist) are members of a dyadic relationship. This relationship rather than individual interactions can influence perceptions and interpersonal dynamics during therapy. For example, the authors of a recent study, comprising 12 general practitioners and 189 patients, conducted an analysis of dyadic relations to assess interpersonal trust in doctor and patient relationships and concluded that relationship and reciprocity effects converged with perception of trust [35]. As we can see in the dyadic interaction (Avatar: Reinforcement, Patient: Self-affirmation), in which a reinforcement assertion is followed by a positive self-assertion, there is a transference of positivity across the members of the dyadic relationship. It has been demonstrated that positive transference can act as a moderator during therapy and can help the therapist in the moderation of the treatment phase and depth [36]. Therefore, by reinforcing the patient, the therapist (as the Avatar) might help the patients in acquiring confidence and learning to affirm themselves. A similar hypothesis can be applied to the reverse dyad (Patient: Self-affirmation, Avatar: Reinforcement) and the related dyad (Patient: Self-appraisal, Avatar: Reinforcement), in which the Avatar might attempt to build on this self-confidence to elicit further self-affirmation.

Similarly, provocation is part of AT in order to mimic the negative experiences of auditory hallucinations experienced as reported by the patients. Provocation is one of the key factors known to provoke anger in human relationships [37]. However, as per the (Avatar: Provocation, Patient: Self-affirmation), the patient’s common reaction in AT is to respond expressing self-affirmation. Patients suffering from schizophrenia are reported to have difficulties in adequately identifying social cues [38,39]. Therefore, a provocation expressed by the Avatar could be misinterpreted as an opportunity to affirm or re-affirm a statement regarding their experience with their auditory hallucinations. In the community, patients suffering from auditory hallucinations often adopt safety-seeking behaviors, which maintains their distorted beliefs regarding their hallucinations [40]. In AT, they cannot adopt these safety-seeking behaviors and are in a secured environment to react to their auditory hallucinations, which might explain why self-affirmations are more prevalent. Another hypothesis is that in the AT protocol, the therapist encourages the patient to affirm themselves when the Avatar is talking to them. In that sense, it could indicate that the patient is responding adequately to the coaching of the therapist.

For patients suffering from TRS, an immersive environment such as the virtual reality environment provided by AT helps the therapists in understanding their relationship with their auditory hallucinations in an in vivo setting [41]. This provides insight as to how they may react under specific conditions. The dyadic interaction (Avatar: Coping mechanisms, Patient: Prevention) enables the Avatar to directly elicit prevention strategies from the patient to better understand their cognitive processes in a particular context. Self-generated coping strategies are important in the treatment of psychotic symptoms and these strategies can be assessed through this particular dyad, which is why it might be as prevalent in AT [42].

It is interesting to note that only five dyadic interactions were identified as having a mean frequency above 10 across the participants. This can be hypothesized to be linked by the interpersonal differences across the patients and their own personal experiences with their auditory hallucinations. The therapist personal traits and disposition can affect the therapeutic alliance whereas the patient’s own personality traits can also affect it [43]. With the current shift towards precision medicine, it is not surprising to see personalized adaptation of psychotherapy for specific patients [44]. AT is no exception to this, as patients presenting with TRS can be widely heterogenous, and therefore, the interpersonal dynamics can yield multiple different dyadic interactions.

### Limitations

This remains an exploratory content analysis and cannot be generalized outside the scope of AT. Considering that two therapists were involved in conducting AT for these patients, the interactions cannot be interpreted as generalizable over all therapists, although there was a consistency observed between all participants and the dyads that emerged. While the amount of verbatims that was analyzed is substantial, the automated annotation has a recurrent bias in its classification, as themes that performed poorly during training due to a lack of data in the initial AT dataset will be misidentified. However, it can be hypothesized that themes that do not occur frequently in AT immersive sessions would signify a smaller mean frequency of apparitions for dyadic interactions that would encompass those themes.

## 5. Conclusions

In conclusion, the main goal of this study was to identify the dyadic interactions between the patient and the Avatar that are the most prevalent in AT for patient’s suffering from treatment-resistant schizophrenia. The automated annotation of the verbatims and the content analysis of the identified dyadic interactions highlighted five major dyads in AT: (Avatar: Reinforcement, Patient: Self-affirmation), (Avatar: Provocation, Patient: Self-affirmation), (Avatar: Coping mechanisms, Patient: Prevention), (Patient: Self-affirmation, Avatar: Reinforcement), and (Patient: Self-appraisal, Avatar: Reinforcement). These dyads provide a first insight as to the interpersonal dynamics occurring in AT. Future studies on the implication of such dyadic interactions with the therapeutic outcome of AT should be conducted considering the importance of dyadic relationships in psychotherapy.

## Figures and Tables

**Figure 1 jcm-12-02299-f001:**
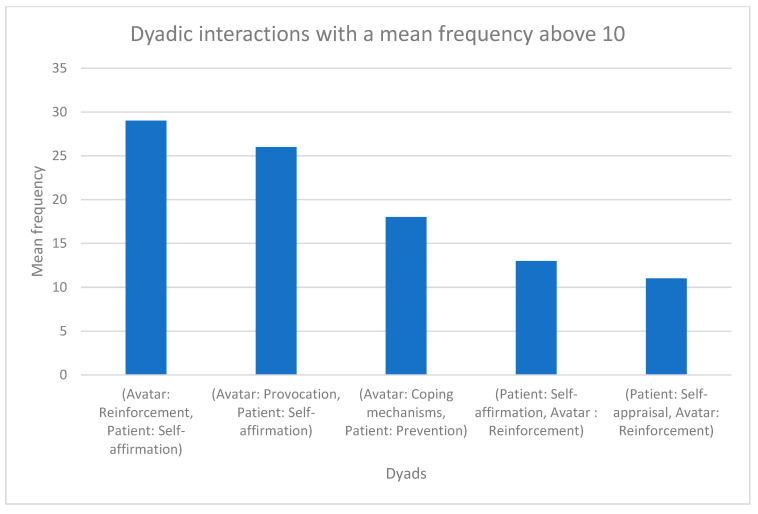
Dyadic interactions and their mean frequencies. Only dyads with a mean frequency per participants above 10 over the entire AT were presented.

**Table 1 jcm-12-02299-t001:** Avatar interactions themes as per Beaudoin et al. 2021.

Avatar Themes	Examples
Accusations	“You are responsible for this.”
Omnipotence	“I am the best.”
Beliefs	“I believe that you are ill.’’
Active listening, empathy	“There is no rush, take your time.”
Incitements, orders	“You should hit yourself.”
Coping mechanisms	“Why are you not happy when I insult you?”
Threats	“I will kill you.”
Negative emotions	“It’s difficult for me to realize that.”
Self-perceptions	“I see myself as worthless.”
Positive emotions	“I am feeling great.”
Provocation	“Try me.”
Reconciliation	“Should we stop arguing?”
Reinforcement	“You should do this again.”

**Table 2 jcm-12-02299-t002:** Patient interactions’ themes as per Beaudoin et al. 2021.

Patient Themes	Examples
Approbation	“You are right”
Self-deprecation	“I can’t do this.”
Self-appraisal	“I am a nice person.”
Other beliefs	“You are the one controlling me”
Counterattack	“You are the one who did this, not me!”
Maliciousness of the voice	“You are trying to make this hard for everyone.”
Negative	“It is not easy.”
Negation	“I did not do this.”
Omnipotence	“I am everywhere.”
Disappearance of the voice	“Please vanish!”
Positive	‘’I am feeling great.”
Prevention	“I will try not pay attention to you.”
Reconciliation of the voice	“Let’s be friends.”
Self-affirmation	“I can do this.”

**Table 3 jcm-12-02299-t003:** Classification performances for the automated annotation of the AT verbatims.

Avatar Themes	Precision	Recall	f1-Score	#Interactions Tested
Accusations	0.74	0.66	0.70	35
Omnipotence	0.77	0.94	0.85	18
Beliefs	0.75	0.69	0.72	26
Active listening, empathy	0.70	0.82	0.76	17
Incitements, orders	0.78	0.70	0.74	10
Coping mechanisms	0.96	0.88	0.92	25
Threats	1.00	0.86	0.92	7
Negative emotions	0.79	0.92	0.85	12
Self-perceptions	0.60	0.71	0.65	17
Positive emotions	0.91	0.62	0.74	16
Provocation	0.67	0.62	0.65	16
Reconciliation	0.64	0.69	0.67	13
Reinforcement	0.73	0.84	0.78	19
Accuracy			0.76	231
Weighted average	0.77	0.76	0.76	231
**Patient Themes**	**Precision**	**Recall**	**f1-score**	**#Interactions tested**
Approbation	0.31	0.31	0.31	13
Self-deprecation	0.53	0.67	0.59	12
Self-appraisal	0.86	0.59	0.70	32
Other beliefs	0.52	0.65	0.58	17
Counterattack	0.70	0.54	0.61	26
Maliciousness of the voice	0.62	0.71	0.67	14
Negative	0.72	0.64	0.68	36
Negation	0.79	0.73	0.76	30
Omnipotence	0.27	0.60	0.37	10
Disappearance of the voice	0.78	0.64	0.70	22
Positive	0.83	0.88	0.86	17
Prevention	0.86	0.59	0.70	32
Reconciliation of the voice	0.30	1.00	0.46	3
Self-affirmation	0.46	0.62	0.53	21
Accuracy			0.64	285
Weighted average	0.69	0.64	0.65	285

**Table 4 jcm-12-02299-t004:** Patients’ characteristics.

Characteristics	Value (*n* = 32)
Sex (male, female)	24.8
Age (mean in years)	42.6 ± 11.0
Education (mean in years)	13.6 ± 3.0
Ethnicity (Caucasian, others)	93.4%, 6.6%
% on Clozapine	40.0%

## Data Availability

The data presented in this study are available on request from the corresponding author. The data are not publicly available due to patients’ privacy.

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
