# Peer review of "Dyadic Interactions of Treatment-Resistant Schizophrenia Patients Having Followed Virtual Reality Therapy: A Content Analysis"

_jcm, 2023, doi:10.3390/jcm12062299_

Round 1

Reviewer 1 Report

Overall, I think this is an interesting exploration of a novel technique for addressing AH in TRS. However, my primary concern is that the findings are limited and may be confounded by the circumstances of this therapeutic approach. Secondly, there can be a stronger argument at the beginning of the manuscript to justify the need for such review of thematic interactions. I have detailed my notes below:

1. Limited findings potentially biased by the therapeutic approach: The authors appropriately note in the limitations that the findings are exploratory and limited within AT, however, I would expect that this may also be a function of the therapists performing the AT. As I understand the process - the therapist is both acting as the Avatar and coaching the patient in it's responses. Were there any findings on the prevalence of certain dyadic responses analyzed by therapist administering the AT? I would expect that certain therapists take specific approaches more often than others. If all 32 patients were treated by the same therapist - how generalizable are these dyadic responses when implemented by another therapist? On page 8, the paragraph beginning "Similarly, provocation is part of AT..." acknowledges that the therapist plays a role in how patients respond within these sessions - effectively driving both parts of these dyadic responses (Lines 288 - 290; "Another hypothesis...coaching of the therapist"). I do not think this negates the findings, but is a significant point that should be addressed. My following comment may be outside the scope of the current paper, but following this line of thought, I am curious as to the impact of these specific dyadic responses on outcome - is there a way to examine prevalence of these dyads and the relationship to improvement on AH? 

2. Improvements to the introduction/justification for the current exploratory review: The authors note in the introduction that AT may relate to transference and countertransference , but do not make a clear connection beyond that therapists typically should not provoke or ignore their patient within CBT. It may be beneficial to include within the introduction some comments that in AT, the therapists goal is to provoke the patient to encourage them to gain power over their AHs. Some statements from the conclusion that more clearly define this relationship are on pg 7 lines 260-262 ("During AT, the patients... interpersonal dynamics during therapy") and 268-270 ("there is a transference...treatment phase and depth.")

Author Response

Please see the attached comments. Thank you !

Reviewer 2 Report

In this manuscript, the author introduced a newly developed evidence regarding the outcomes of Avatar Therapy for treatment-resistant schizophrenia. It sounds very good, for its uniqueness and availablity for clinicians engaging in psychotherapy.

In this study, the author combined the data of participants of two different clinical trial; one is a pilot study, and the other is a controlled trial. Mixture of the data itself is acceptable for this kind of analysis presented in this paper. But, are there any difference between the two previous studies regarding the method of each study? If so, combining the results of each study can be inappropreate. The author should clarify this issue in the method section.

Author Response

(The authors gave the same response as above.)

Reviewer 3 Report

I suggest you plan a research with a larger sample. Good choice of topic and methodological approach

Author Response

(The authors gave the same response as above.)
